# Spherical-Nested Diffusion Model for Panoramic Image Outpainting

**Xiancheng Sun**[1]   **Senmao Ma**[1]   **Shengxi Li**[1 2]   **Mai Xu**[1]   **Jingyuan Xia**[3]   **Lai Jiang**[1 2]   **Xin Deng**[1 2]   **Jiali Wang**[4]

## Abstract

Panoramic image outpainting acts as a pivotal role in immersive content generation, allowing for seamless restoration and completion of panoramic content. Given the fact that the majority of generative outpainting solutions operates on planar images, existing methods for panoramic images address the sphere nature by soft regularisation during the end-to-end learning, which still fails to fully exploit the spherical content. In this paper, we set out the first attempt to impose the sphere nature in the design of diffusion model, such that the panoramic format is intrinsically ensured during the learning procedure, named as spherical-nested diffusion (SpND) model. This is achieved by employing spherical noise in the diffusion process to address the structural prior, together with a newly proposed spherical deformable convolution (SDC) module to intrinsically learn the panoramic knowledge. Upon this, the proposed method is effectively integrated into a pre-trained diffusion model, outperforming existing state-of-the-art methods for panoramic image outpainting. In particular, our SpND method reduces the FID values by more than 50% against the state-of-the-art PanoDiffusion method. Codes are publicly available at https://github.com/chronos123/SpND.

## 1. Introduction

Most recently, panoramic images have received increasing research efforts due to their critical roles in applications such as virtual reality (VR) (Wan Abd Arif et al., 2009; Wang,

---
[1]Department of Electronic Information Engineering, Beihang University, Beijing, China [2]State Key Laboratory of Virtual Reality Technology and Systems, Beihang University, Beijing, China [3]Department of Electronic Science and Technology, National University of Defense Technology, Changsha, China, Location, Country [4]Cainiao Technology, Hangzhou, China. Correspondence to: Shengxi Li <LiShengxi@buaa.edu.cn>.

*Proceedings of the 42^{nd} International Conference on Machine Learning*, Vancouver, Canada. PMLR 267, 2025. Copyright 2025 by the author(s).

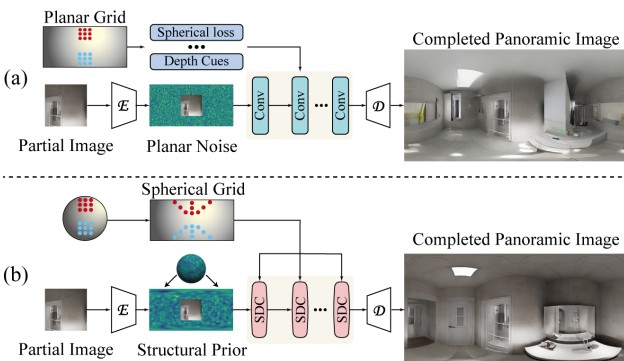

*Figure 1.* Comparison between the proposed and existing methods for panoramic image outpainting. (a) Existing methods learn panoramic content by soft regularisations, in a *macro way*, thus outpainting irregular 3D arrangement and artefacts. (b) The proposed method incorporates the sphere nature within the design, thus in a *micro way*, which is able to outpaint realistic and high-quality panoramic images. Note that $E$ denotes for the encoder and $\mathcal{D}$ denotes for the decoder.

2019; Zhang & Kou, 2022), augmented reality (AR) (Lee & Tsai, 2015; Eiris et al., 2018), and autonomous driving (Huang et al., 2018; Yurtsever et al., 2020; Gao et al., 2022). Although existing methods (Rombach et al., 2022) are capable of generating high-quality content, they are designed for planar images and fail to generalise well to panoramic images, due to the spherical distortion introduced by the equirectangular projection (ERP). Consequently, the generation of panoramic images has emerged as a pivotal area of research, drawing extensive research interest seeking to produce high-quality panoramic content. We also notice several text-to-panorama generation methods, including PanFusion (Zhang et al., 2024) and MVDiffusion (Tang et al., 2023) by incorporating attention mechanism to partially address panoramic cues. They still fail to address spatial continuity and contextual coherence for panoramic image outpainting.

As one of the main-stream tasks of panoramic image generation, panoramic image outpainting has been pioneered by works (Akimoto et al., 2019; Sumantri & Park, 2020), which are able to complete scenes with the $360° \times 180°$ field-of-view (FOV). To ensure spherical continuity of the generated edges, one of the core characteristics of panoramic images, existing solutions include the usage of circular padding technique (Hara et al., 2021; Akimoto et al., 2022) and rotation

within the latent space (Wang et al., 2023a). A further crucial problem is the accurate capturing against the panoramic content, which is distorted by ERP during the training. Existing methods relieve this by employing the panoramic-guided loss (Wang et al., 2024), extra depth modality cues (Wu et al., 2024b), or unconstrained deformable convolution operations (Wu et al., 2024a), which essentially impose soft regularisations in an *macro way* to guide the overall training procedure. Generative models trained in an end-to-end style by the soft regularisations, however, may not be able to operate as expected, thus failing to fully exploit the spherical geometry and panoramic content as illustrated in Figure 1.

In this paper, we set out the first attempt to outpaint panoramic images by incorporating their intrinsic sphere nature in an *micro way*, i.e., by the sophisticated design of generative models. More specifically, we first propose to sample identically and independently distributed (*i.i.d.*) spherical noise on the 3D surface in the diffusion model, corresponding to the sphere nature of panoramic images. To the best of our knowledge, all the existing methods sample noise on the planar latent space. Though, the *i.i.d.* spherical noise essentially results into non-*i.i.d.* planar noise after the ERP projection; this ensures that our method suits well in the panoramic image outpainting task. More importantly, given the fact that an irregular grid is required to maintain the regular grid on the sphere, we propose to adaptively perceive the receptive field on the sphere, instead of on the plane, in which the spherical deformable convolution (SDC) layer is developed in particular for the panoramic content. Our SDC layer introduces a fundamentally novel convolution operation to mitigate panoramic deformation, which is distinct from existing strategies such as feature re-organisation (Shen et al., 2022) or sphere-guided loss (de La Garanderie et al., 2018; Jaus et al., 2023) that are employed for panoramic depth estimation and segmentation tasks. Built upon the SDC layers, we propose our spherical-nested diffusion (SpND) model, based on a newly developed circular mask encoder (CME) for the consistency across edges. The experimental results have verified the significantly superior performances of our SpND model. Our contributions are three-fold:

- We seamlessly incorporate the spherical noise by investigating ERP distortion when training diffusion models, such that the structural prior of panoramic images can be well accommodated.

- We propose the SDC layer that is the first successful attempt to satisfy the intrinsic sphere nature within generative model architectures, with adaptive and consistent receptive fields.

- We develop the SpND model by incorporating the spherical noise and SDC layer as fundamental modules,

accomplished by the CME to ensure the high-quality panoramic image outpainting.

## 2. Related Works

**Image Outpainting.** The image outpainting task aims to fill unknown regions beyond the boundaries of given content. Based on advancements of generative models, such as variational autoencoders (VAEs) (Kingma & Welling, 2014) and generative adversarial networks (GANs) (Goodfellow et al., 2014), a series of image outpainting methods has been developed (Pathak et al., 2016; Iizuka et al., 2017; Yu et al., 2018; Zheng et al., 2019; Wang et al., 2019; K. & Ali, 2020; Zhao et al., 2021). The generative models allow for outpainting semantically meaningful content when completing images based on a provided portion from an overall image. The most recent advancements in diffusion model (Rombach et al., 2022) with the transformer architecture (Vaswani et al., 2017) have further improved completed images with consistent structure and semantics (Dosovitskiy et al., 2021; Esser et al., 2021; Wan et al., 2021; Zheng et al., 2022; Yao et al., 2022; Lugmayr et al., 2022; Xie et al., 2023; Wang et al., 2023b; Li et al., 2024). However, regarding panoramic images, these methods overlook the unique characteristics of panoramic images, including omnidirectional continuity across edges and spherical distortion.

**Panoramic Image Outpainting.** Different from planar images with a narrow FOV, panoramic images with a $360° \times 180°$ FOV exhibit spherical distortion due to the ERP projection. Consequently, objects in the ERP format undergo significant distortion, particularly for those located near the top and bottom regions of the image. Image completion methods for panoramic images are thus required to address the spherical distortion inherent in their structure and ensure the preservation of omnidirectional continuity. Early studies (Sumantri & Park, 2020; Akimoto et al., 2019; Lu et al., 2021) primarily focus on ensuring omnidirectional continuity by adopting convolution operation with the circular padding strategy. To address the spherical distortion, there are strategies such as using extra depth information (Oh et al., 2022), adding projection loss (Somanath & Kurz, 2021), and leveraging scene-symmetry properties (Hara et al., 2021). In order to generate visually pleasant content, Omnidreamer (Akimoto et al., 2022) employs transformer-based sampling strategy to implicitly learn the spherical structure. To achieve a smooth transition between the real and generated content, Cylin-Painting (Liao et al., 2024) incoporates learnable positional encoding cues with existing features that are still processed by standard 2D convolution. Leveraging the powerful diffusion model, PanoDiff (Wang et al., 2023a) employs the ControlNet (Zhang et al., 2023) with the positions as an additional input and also uses spherical loss to regularise panoramic content. To further accomodate intricate visual details, AOG-Net (Lu et al., 2024)

proposes an autoregressive pipeline for panoramic image outpainting, utilising feature remapping for panoramic content. On the other hand, PanoDiffusion (Wu et al., 2024b) introduces the depth modality during training to enhance panoramic awareness. Unfortunately, the above methods dominantly address the distortion by soft regularisations, from the perspective of guiding the overall end-to-end training, i.e., in a *macro way*. This may result in suboptimal performance since the macro constraints may not fully contribute to learning the panoramic content as expected.

## 3. Methodology

### 3.1. Spherical Noise for Structural Prior

The noise plays an essential role that relates to the randomness of outpainting, which benefits the overall performance by considering the panoramic structural prior. More specifically, since panoramic images are commonly projected into the ERP planar format (Ye et al., 2018) for processing, we aim to build an explicit structural prior knowledge for the ERP format to guide the panoramic image outpainting process. Given the fact that images in the ERP format exhibit higher pixel density near the equator and lower pixel density near the poles, modelling the coherence of panoramic images using *i.i.d.* Gaussian noise, the *de facto* choice for planar images, is mismatched with the intrinsic characteristics of panoramic ERP format. Instead, by modelling panoramic images as pixels uniformly distributed over the surface of a sphere, we are able to capture pixel coherence in the ERP format by sampling *i.i.d.* noise uniformly distributed in 3D space and projecting it into the ERP format at given resolutions.

Firstly, we take a analysis on the spherical property of the ERP format. We denote $u, v$ as the coordinates in the ERP format and $\theta, \phi$ are the spherical coordinates on the surface of the sphere, two widely used coordinates for panoramic images. Correspondingly, the transformation from spherical format to the ERP planar can be formulated by $u = \theta w / 2\pi$, $v = (\phi + \frac{\pi}{2})h/\pi$, where $w$ and $h$ are the width and height of the image, respectively. Also, the spherical coordinates $(\theta, \phi)$ are defined by $x = \cos\phi\cos\theta, y = \cos\phi\sin\theta, z = \sin\phi$ where $x, y, z$ are the Cartesian coordinates in 3D space. Then, we assume that there are $m_w$ sampling points per row in the ERP format. The actual geodesic distance $d_v$ between two adjacent sampling points on the spherical surface, at height $v$ in the ERP format, is given by

$$d_v = \frac{2\pi}{m_w} \cos\left(\frac{\pi v}{h} - \frac{\pi}{2}\right). \tag{1}$$

Therefore, given the points at poles ($v = 0$ and $v = h$), $d_v = 0$ means that all points essentially correspond to a single point in 3D space. We summarise this property as the sphere nature of the ERP format.

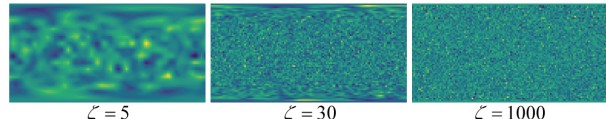

$\zeta = 5$      $\zeta = 30$      $\zeta = 1000$

*Figure 2.* Illustration of the *i.i.d.* spherical noise which exhibits non-*i.i.d.* nature after the ERP projection. We illustrate examples with different sampling densities $\zeta$ on the 3D spherical surface at a resolution of $128 \times 64$.

Based on the above analysis, we propose to sample noise in 3D space and transform it into the ERP format to capture the intrinsic structural prior. The uniform spherical noise is sampled by adopting sampling density $\zeta$ per unit length in 3D space. Accordingly, the number of sampling points $m_\phi$ at latitude $\phi$ on the spherical surface can be formulated by

$$m_\phi = \lceil \zeta \cdot 2\pi \cos\phi \rceil \tag{2}$$

where $2\pi\cos\phi$ denotes the geodesic length and $\lceil \cdot \rceil$ is the ceiling function. Note that the number of sampling point at poles is 1. $\zeta$ essentially controls the sampling density of the *i.i.d.* Gaussian noise on 3D spherical surface. Therefore, given $m_w$ as the fixed number of points for a row in the ERP format, for latitudes where $m_\phi < m_w$, one sampling point on the spherical surface corresponds to multiple points within a single row in the ERP format. This one-to-many mapping inherently introduces spatial coherence in the ERP domain. As a result, *i.i.d.* noise in 3D space is no longer *i.i.d.* when projected into the ERP format. In contrast, for latitudes where $m_\phi \geq m_w$, the *i.i.d.* property within a row in the ERP representation can be preserved. Both the sampling density $\zeta$ and the resolution, which determines the number of sampling points $m_w$, play an important role in modelling the sphere nature of the ERP format. Since the resolution of feature is set as $128 \times 64$ in our model, we further illustrate the non-*i.i.d.* noise (aka. structural prior) with different $\zeta$ in Figure 2. We also illustrate detailed analysis in Appendix A, by choosing $\zeta = 30$ for our SpND model.

### 3.2. Spherical Deformable Convolution

Generally speaking, the deformable convolution (Dai et al., 2017) decomposes the standard convolution operation into two steps: 1) Sampling certain points from the input feature $\mathbf{F}_{\text{in}}$ using a regular grid $\mathcal{R}$; 2) Summing the sampled points weighted by $\mathbf{W}$. In the standard convolution, the grid

$$\mathcal{R} = \{(-1, -1), (-1, 0), (-1, 1), (0, -1), \cdots, (0, 1), (1, 1)\} \tag{3}$$

defines a $3 \times 3$ kernel with stride 1 and determines the receptive field. For each location $\mathbf{p}_0$ on the output feature $\mathbf{F}_{\text{out}}$, the deformable convolution can be formulated by

$$\mathbf{F}_{\text{out}}(\mathbf{p}_0) = \sum_{\mathbf{p}_n \in \mathcal{R}} \mathbf{W}(\mathbf{p}_n) \cdot \mathbf{F}_{\text{in}}(\mathbf{p}_0 + \mathbf{p}_n + \Delta\mathbf{p}_n), \tag{4}$$

where $\mathbf{p}_n$ enumerates the locations in $\mathcal{R}$ and $\{\Delta\mathbf{p}_n|n = 1,\cdots,N\}$ denote the learnable offsets, where $N = |\mathcal{R}|$.

Although deformable convolution improves the flexibility of the convolution operation, it still operates as the planar convolution, thus falling short in highly complex geometric scenes, such as spherical data of panoramic images. This issue stems from the fact that directly applying deformable convolution to panoramic images results in suboptimal initialization, as it relies on the regular grid $\mathcal{R}$ as the initial grid (Dai et al., 2017); this misaligns with the spherical structure inherited from panoramic images.

To address the mismatch, we propose the novel SDC layer that leverages the spherical gird, while integrating dynamic adaptability of deformable convolution for precise local feature aggregation. More specifically, the spherical neighbourhood of a kernel at position $\mathbf{p}_0$ can be defined as a spherical grid $\mathcal{R}^s$ as introduced in (Coors et al., 2018), using gnomonic projection (Frederick Pearson, 1990) and a tangent plane centred at $\mathbf{p}_0$ on sphere. The coordinate for $\mathbf{p}_0 = (\theta_0, \phi_0)$ is defined by its latitude $\phi_0 \in \left[-\frac{\pi}{2}, \frac{\pi}{2}\right]$ and longitude $\theta_0 \in [-\pi, \pi]$. Recall that $w$ and $h$ are the width and height of the image in ERP format (Ye et al., 2018). Then, the $3 \times 3$ grid $\mathcal{R}^t$ on the tangent plane (Coors et al., 2018) is reformulated from (3) as

$$\begin{aligned} \mathcal{R}^t = \{&(-\tan\Delta_\theta, -\sec\Delta_\theta\tan\Delta_\phi), (-\tan\Delta_\theta, 0), \\ &(-\tan\Delta_\theta, \sec\Delta_\theta\tan\Delta_\phi), (0, -\tan\Delta_\phi), \\ &\cdots, (0, \tan\Delta_\phi), (\tan\Delta_\theta, \sec\Delta_\theta\tan\Delta_\phi)\}, \end{aligned} \quad (5)$$

where $\Delta_\phi = \frac{\pi}{h}$ and $\Delta_\theta = \frac{2\pi}{w}$ denote unit steps on the tangent plane. By representing the elements within $\mathcal{R}^t$ as $\mathcal{R}^t = \{(\Delta i_n, \Delta j_n)|n = 1,\cdots,N\}$, we can obtain the spherical gird $\mathcal{R}^s = \{(\theta^n, \phi^n)|n = 1,\cdots,N\}$ centred at position $\mathbf{p}_0$ through

$$\begin{aligned} \theta^n &= \theta_0 + \tan^{-1}\left(\frac{\Delta i_n}{\cos\theta_0 - \Delta j_n \sin\theta_0}\right), \\ \phi^n &= \sin^{-1}\left(\frac{1}{1+\Delta d}(\sin\phi_0 + \Delta j_n \cos\phi_0)\right), \end{aligned} \quad (6)$$

where $\Delta d = \sqrt{(\Delta i_n)^2 + (\Delta j_n)^2}$ (Coors et al., 2018).

Therefore, the spherical grid $\mathcal{R}^s$ defines the $3 \times 3$ neighbourhood around position $\mathbf{p}_0$ on the sphere in our SDC layers. While a kernel size of 3 is used here as an example, the kernel size can be arbitrarily chosen in practice.

Furthermore, since the spherical grid serves as the initial grid for the SDC layer, the usage of unconstrained learnable offsets $\Delta\mathbf{p}_n$ may deteriorate the deformable convolution to neglect the spherical prior knowledge. To address this issue, we impose constraints on the offsets to ensure that the SDC layer consistently integrates the spherical prior knowledge. For each location $\mathbf{p}_0$ on the output feature $\mathbf{F}_{\text{out}}$ and the

input feature $\mathbf{F}_{\text{in}}$, the SDC layer can be formulated as

$$\mathbf{F}_{\text{out}}(\mathbf{p}_0) = \sum_{\tilde{\mathbf{p}}_n \in \mathcal{R}^s} \mathbf{W}(\tilde{\mathbf{p}}_n) \cdot \mathbf{F}_{\text{in}}(\mathbf{p}_0 + \tilde{\mathbf{p}}_n + \Delta\mathbf{p}_n), \quad (7)$$

where $\tilde{\mathbf{p}}_n$ enumerates the locations in the spherical grid $\mathcal{R}^s$, $\mathbf{W}$ denotes the learnable weights of the kernel, and the constraint for the learnable offsets is formulated by $|\Delta\mathbf{p}_n| \leq \sigma \cdot h$ where $h$ denotes the height and $\sigma$ is a hyperparameter which defines the ratio against the overall height.

### 3.3. Spherical-Nested Diffusion Model

Considering the sphere nature of panoramic images, our SpND model achieves panoramic image outpainting by integrating the structural prior and the newly proposed SDC layer. The overall architecture of our SpND model is illustrated in Figure 3. We further develop the CME module to preserve omnidirectional continuity and the spherical structure, thereby significantly enhancing the quality of the generated content. More specifically, our CME is developed to encode the masked input image, which contains the known portion within panoramic images. Then, the trainable CME module employs $3 \times 3$ circular convolution layers with the sigmoid linear unit (SiLU) activation function (Elfwing et al., 2018), to transform the 3-channel masked image into the 256-channel feature, with an overall downsampling ratio of 8. Subsequently, the 256-channel latent feature is fed into the prior injection layer, which aggregates the structural prior for the ERP format. This process produces an intermediate output containing both ERP structural information and image completion details. We denote $\mathbf{X}_{\mathbf{m}}$ as the masked input and $\mathbf{P}_{\text{ERP}}$ as the ERP structural prior. The prior injection layer operats as follows,

$$\mathbf{F}_{\text{pri}} = \mathbf{m} \odot \text{CME}[\mathbf{X}_{\mathbf{m}}] + (1 - \mathbf{m}) \odot \mathbf{P}_{\text{ERP}} \quad (8)$$

where $\mathbf{F}_{\text{pri}}$ denotes the aggregated feature, $\mathbf{m}$ is the mask, $\odot$ represents the element-wise production, and $\text{CME}[\cdot]$ means the encode process of the CME module. Note that the mask $\mathbf{m}$ is provided by the input.

The spherical net is then established based on the spherical-nested blocks (SpNBs), which is introduced to extract panoramic features that guide the pre-trained diffusion model for panorama outpainting. More specifically, each SpNB incorporates an SDC layer to effectively encode spherical information, catering for the sphere nature with panoramic images. Furthermore, a vanilla deformable convolution layer is employed to bridge the spherical feature and the pre-trained diffusion model. This additional layer enhances compatibility with the pre-trained diffusion model, allowing for effective integration and improved performances.

Aligned with the architecture of ControlNet (Zhang et al., 2023), we adopt the SiLU activation function (Elfwing et al.,

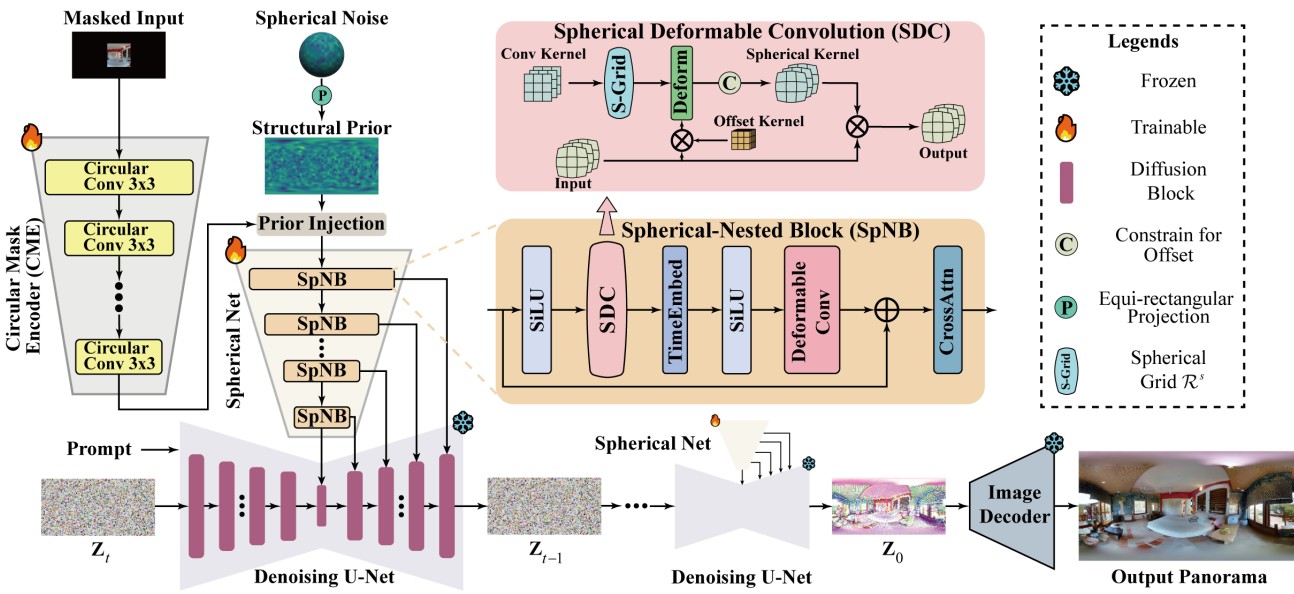

*Figure 3.* The overall architecture of our proposed SpND model. The input of our SpND model is the masked images, which are first encoded by our circular mask encoder (CME) module. Subsequently, the prior injection layer integrates structural prior knowledge tailored for the ERP format into the encoded masked features. These features are subsequently refined by the spherical net, composed of SpNBs, which incorporate spherical prior knowledge inherent in the SDC layers of the SpNBs. The refined features then guide a pre-trained diffusion model to perform panoramic image outpainting using the repaint technique.

2018) to process features encapsulating the spherical property, combined with information from the original image, and sequentially integrate into the middle and output layers of the diffusion model. Assume that $f_{\mathrm{SD}}$ denotes the pre-trained diffusion model and $f_{\mathrm{PDN}}$ denotes the spherical net. The process can be formulated as

$$\epsilon_{\psi} = f_{\mathrm{SD}}[\mathbf{Z}_t, \mathbf{T}, f_{\mathrm{PDN}}[\mathbf{F}_{\mathrm{pri}}, \mathbf{T}]], \qquad (9)$$

where $\mathbf{T}$ is the embedding of the input prompt and $\epsilon_{\psi}$ is the output of our SpND model. Note that $\psi$ denotes the learnable parameters. Furthermore, the final output $\epsilon_{\psi}$ is decoded by the image decoder to get the completed panoramic image. Note that the image decoder is a pre-trained VAE model (Rombach et al., 2022). Moreover, the text embedding $\mathbf{T}$ is derived by encoding the text prompt using the widely adopted CLIP model (Radford et al., 2021).

During the training process, the pre-trained diffusion model and the VAE for encoding panoramic images are frozen, whilst the CME and the spherical net are trainable, as illustrated in Figure 3. We follow the standard process (Rombach et al., 2022) to train the conditional diffusion model via the default diffusion loss, which can be formulated by

$$\mathcal{L} = \mathbb{E}_{t, \mathbf{z}_0, \epsilon}[||\epsilon - \epsilon_{\psi}[\mathbf{Z}_t, t, \mathbf{T}]||_2^2], \qquad (10)$$

where $\epsilon$ is the Gaussian noise, $t$ denotes the time, and $\mathbf{Z}_t$ is the latent feature in the pre-trained diffusion model.

During the inference stage, the repaint technique (Lugmayr et al., 2022) is employed in our SpND model, with the

mask aligned to the input, so as to perform the image outpainting task. We provide more details for the repainting technique in Appendix B. Utilizing the repaint technique for the pre-trained diffusion model ensures that the masked area remains consistent with the input, thereby enabling seamless image outpainting without introducing distortion to the provided regions.

## 4. Experimental Results

### 4.1. Experimental Settings

**Datasets.** To evaluate the performance of our SpND model, we employed the widely applied Matterport3D (Chang et al., 2017) and Structured3D (Zheng et al., 2020) dataset for comparison. Similar to (Lin et al., 2019), we obtained 10912 panoramic images with size $1024 \times 512$ for the Matterport3D dataset. A total of $9,820$ images were selected for the training, and all $1,0912$ images were used for evaluation to compute the sufficient statistics. For the Structured3D dataset, we followed the methodology outlined in (Wu et al., 2024b) to obtain 21,133 images, of which 19,019 images were used for training and all 21,133 images were used for evaluation to compute the sufficient statistics. Prompts were generated by the BLIP-2 (Li et al., 2023) model for both datasets.

**Implementation Details.** Our SpND model was trained based on the pre-trained weights of (Zhang et al., 2023). The hyperparameter $\zeta$ was set to 30 to obtain the struc-

*Table 1.* Quantitative comparisons with state-of-the-art methods on Matterport3D and Structured3D. The best results and the second-best results are highlighted in **bold**, underline.

| - | Matterport3D | | | | | Structured3D | | | | |
|---|---|---|---|---|---|---|---|---|---|---|
| Methods | FID ↓ | FID_hori ↓ | Avg ↑ | Pre ↑ | Recall ↑ | FID ↓ | FID_hori ↓ | Avg ↑ | Pre ↑ | Recall ↑ |
| SIG-SS (Hara et al., 2021) | 64.81 | 50.53 | 0.22 | 0.42 | 0.01 | 117.84 | 72.51 | 0.15 | 0.29 | 0.01 |
| TFill (Zheng et al., 2022) | 77.69 | 85.28 | 0.08 | 0.15 | 0.01 | 85.00 | 84.49 | 0.04 | 0.06 | 0.01 |
| OmniDreamer (Akimoto et al., 2022) | 51.69 | 44.44 | 0.31 | 0.45 | 0.16 | 20.63 | 24.94 | 0.35 | 0.51 | 0.19 |
| PanoDiff (Wang et al., 2023a) | 13.45 | 13.11 | 0.50 | 0.24 | 0.75 | 11.11 | 9.75 | 0.53 | 0.36 | **0.69** |
| AOG-Net (Lu et al., 2024) | 38.56 | 44.27 | 0.10 | 0.16 | 0.04 | 85.39 | 98.11 | 0.10 | 0.19 | 0.01 |
| PanoDiffusion (Wu et al., 2024b) | 39.43 | 17.34 | 0.49 | **0.55** | 0.42 | 9.80 | 10.84 | 0.49 | 0.55 | 0.43 |
| SpND (Ours) | 9.08 | 10.64 | 0.62 | 0.52 | 0.72 | 7.37 | 6.05 | 0.62 | 0.62 | 0.62 |
| SpND_prompt (Ours) | **6.67** | **6.58** | **0.64** | 0.52 | **0.76** | **4.60** | **4.75** | **0.64** | **0.66** | 0.62 |

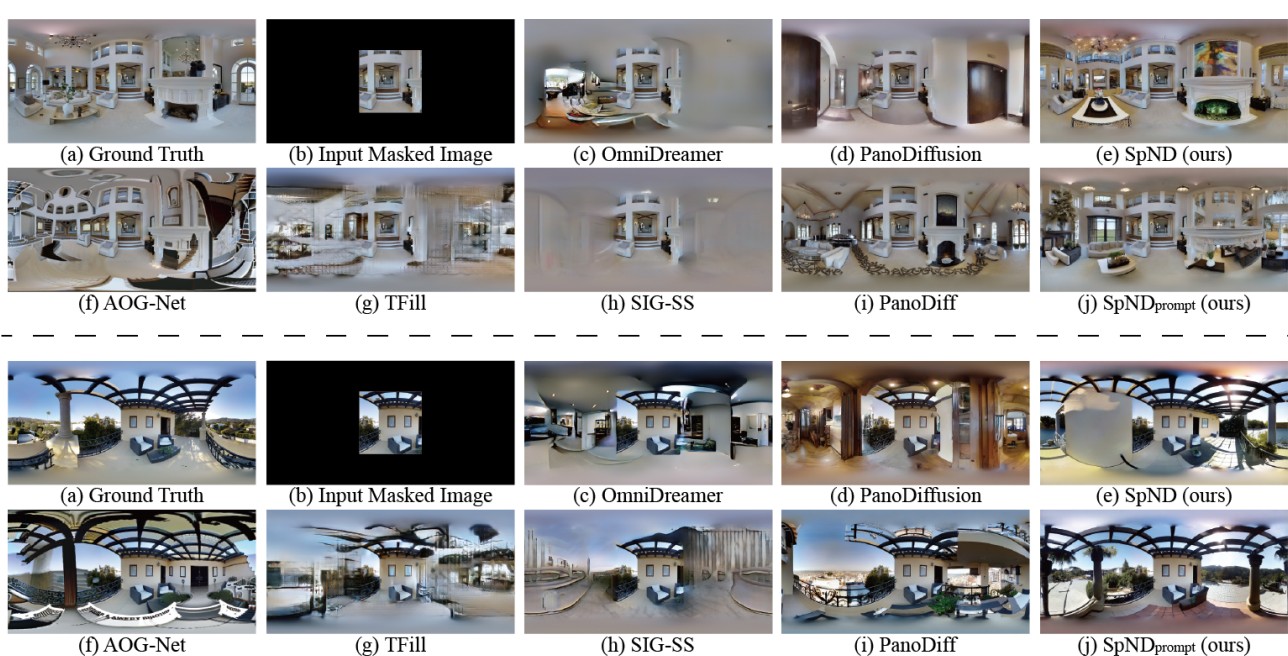

*Figure 4.* Qualitative comparisons of panoramic image outpainting on the Matterport3D dataset. The centre mask is employed to generate the masked input image, and the results are the corresponding completed images obtained by different methods.

tural prior for the ERP format. During training, we used the AdamW optimizer (Loshchilov & Hutter, 2019) with a learning rate of $10^{-5}$ and a batch size of 4. During inference, the classifier-free guidance scale (Ho & Salimans, 2021) was set to 3.0 with a DDIM (Song et al., 2021) step of 50. Additionally, the hyperparameter $\sigma$ in SDC layers was set to 0.04 to constrain the learnable offset. To rigorously evaluate the efficacy of integrating prior knowledge, we utilised the general prompt "*Outpainting*" to validate our approach. This choice was motivated for fair comparisons, by the fact that several baseline methods do not incorporate prompts as part of their input. Furthermore, to demonstrate that performance could potentially be further improved with application-specific prompts, we trained an additional model incorporating varying text prompts, denoted as SpND_prompt.

**Baselines.** For panoramic image outpainting, we mainly compared with the existing state-of-the-art panorama outpainting methods including SIG-SS (Hara et al., 2021), OmniDreamer (Akimoto et al., 2022), PanoDiff (Wang et al., 2023a), AOG-Net (Lu et al., 2024), and PanoDiffusion (Wu et al., 2024b). Additionally, we compared with the image inpainting method TFill (Zheng et al., 2022) following PanoDiffusion (Wu et al., 2024b). We fine-tuned all the baseline methods on the Matterport3D dataset, whilst fine-tuning almost all the baseline methods on the Structured3D datasets (except for the PanoDiffusion model where the pre-trained weights exist on Structured3D) for fair comparison.

**Metrics.** We utilised the commonly adopted Fréchet inception distance (FID) metric to assess the quality of the completed panoramic images. More specifically, we fol-

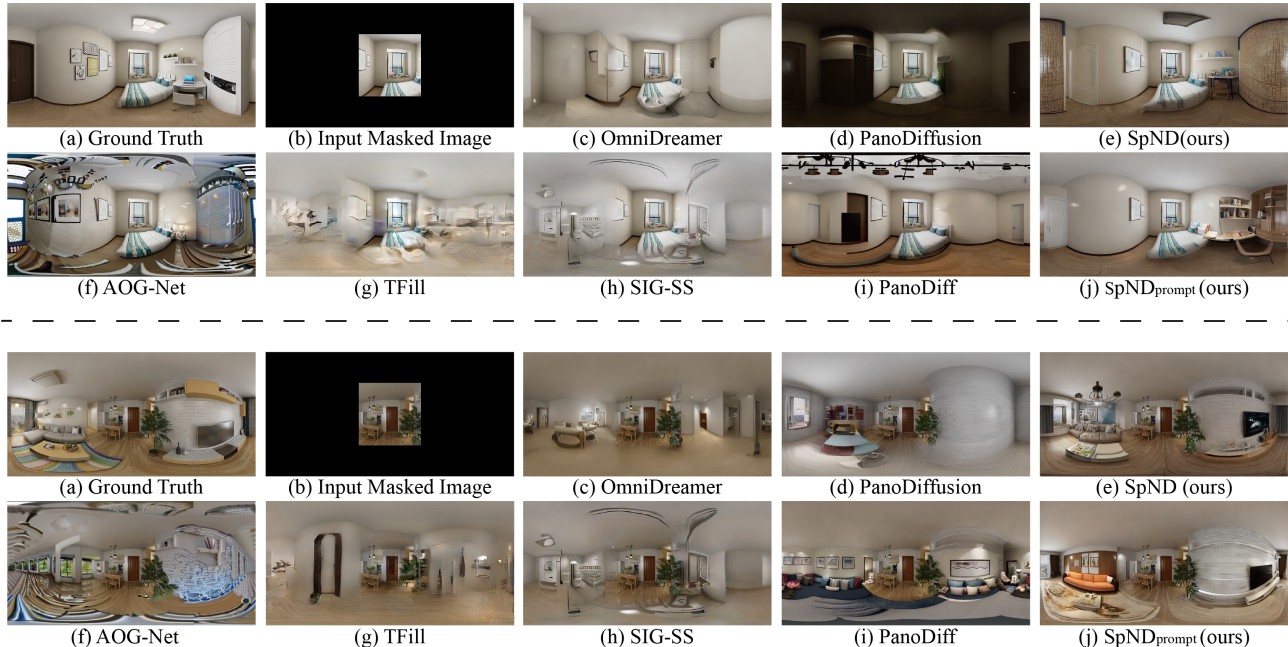

*Figure 5.* Qualitative comparison of panoramic image outpainting on the Structured3D dataset. The centre mask is employed to generate the masked input image, and the results are the corresponding completed images obtained by different methods.

lowed the method in (Wu et al., 2024b) to directly compute the FID score. To further refine the evaluation of image quality for panoramic images, we adopted the approach proposed in (Tang et al., 2023) and computed $FID_{hori}$. This was achieved by calculating the $FID_{hori}$ scores using 8 equally spaced $512 \times 512$ cropped images, extracted from a panoramic image with size $1024 \times 512$. Additionally, the precision (denoted as Pre) and recall metrics (Kynkäänniemi et al., 2019) were also calculated to comprehensively assess outpainting performance. A higher precision value indicates better image quality, while a higher recall value reflects improved diversity in the outpainted images. We average the precision and recall metric (denoted as Avg) to comprehensively evaluate the performance for panoramic image outpainting.

### 4.2. Comparison for Panoramic Image Outpainting

**Quantitative Results.** The widely adopted centre mask was employed in performing the panoramic image outpainting task for all comparing methods. As reported in Table 1, the proposed SpND model consistently achieves the best performance by taking into account both the quality and diversity of the outpainted images. More importantly, with specific prompts as input, the overall outpainting performance can be even better across all metrics, as shown in the last two rows of Table 1. While PanoDiff achieves superior recall for the Structured3D dataset, its FID and the precision are quite limited. In addition, the GAN-based methods including SIG-SS and TFill exhibit extreme low recall scores, indicating that the panoramic outpainting diversity is poor.

**Qualitative Results.** We randomly chose 2 scenes from both dataset and reported the subjective results in Figures 4 and 5. As can be seen from those figures, panoramic images outpainted by our SpND model demonstrate distinct superior quality and optimal alignment with the provided input sections, also achieving the smoothest transitions at the boundary between the real and generated components compared to other methods. Furthermore, with the newly proposed structural prior and SDC layers, our SpND model preserves the spherical structure well and outpaints visually compelling panoramic content. In addition, image quality and semantic continuity can be further improved through the usage of specific prompts in our SpND model.

**Outpainting from View Images.** To further evaluate the performance and the robustness of the proposed SpND model, we conducted panoramic image outpainting experiments using view images. For comparison, we selected the best two existing methods based on their performance in prior experiments and performed the comparison on the Structured3D dataset[1]. As illustrated in Figure 6 and Table 2, our SpND model demonstrates superior performances, achieving the best results both subjectively and objectively.

**Outpainting with Mask near the Pole.** To further evaluate the performance of our method on different masks, we conducted additional experiments on masks located near the poles, whereby the spherical distortion of masks becomes apparent. More specifically, we projected a $256 \times 256$ image

---

[1]We also conducted experiments on a small Martian image dataset using $SpND_{prompt}$, which is detailed in Appendix C.

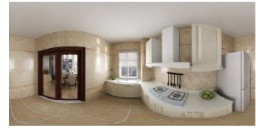 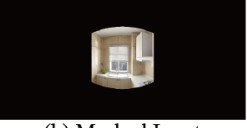 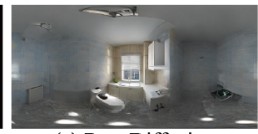 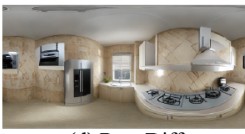 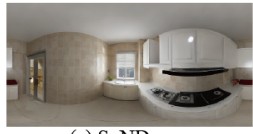

| (a) Ground Truth | (b) Masked Input | (c) PanoDiffusion | (d) PanoDiff | (e) SpND$_{prompt}$ |

*Figure 6.* Qualitative comparison of panoramic image outpainting from view images on the Structured3D dataset. The centre view mask is employed to generate the masked input panoramic view, and the results are the completed images obtained by different methods.

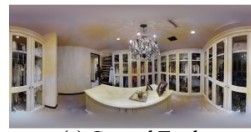 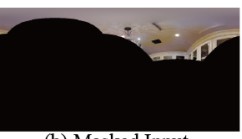 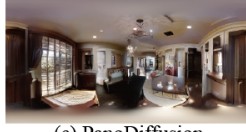 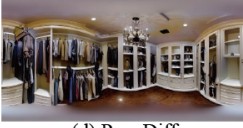 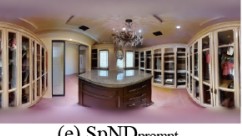

| (a) Ground Truth | (b) Masked Input | (c) PanoDiffusion | (d) PanoDiff | (e) SpND$_{prompt}$ |

*Figure 7.* Qualitative comparison of panoramic image outpainting with mask near the pole on the Matterport3d dataset. The distorted mask near the pole is employed to generate the masked input image, and the results are the completed images obtained by different methods.

*Table 2.* Comparisons for panoramic image outpainting from view images on the Structured3D dataset. The best results are in **bold**.

| Methods | FID ↓ | FID$_{hori}$ ↓ | Avg ↑ | Pre ↑ | Recall ↑ |
|---|---|---|---|---|---|
| PanoDiffusion | 18.06 | 17.64 | 0.36 | 0.38 | 0.34 |
| PanoDiff | 9.20 | 8.19 | 0.54 | 0.40 | **0.68** |
| SpND$_{prompt}$ (ours) | **4.23** | **3.98** | **0.64** | **0.66** | 0.61 |

*Table 3.* Comparisons for panoramic image outpainting with mask near the pole on Matterport3D. The best results are in **bold**.

| Methods | FID ↓ | FID$_{hori}$ ↓ | Avg ↑ | Pre ↑ | Recall ↑ |
|---|---|---|---|---|---|
| PanoDiffusion | 15.92 | 17.76 | 0.48 | **0.54** | 0.42 |
| PanoDiff | 13.97 | 14.91 | 0.48 | 0.30 | 0.65 |
| SpND$_{prompt}$ (ours) | **8.49** | **8.84** | **0.57** | 0.44 | **0.69** |

onto the ERP format at the viewpoint $\theta = 90°, \phi = 60°$, to obtain a highly-distorted mask near the pole. We then re-trained our SpND method, along with the best two baseline methods (i.e., PanoDiff and PanoDiffusion) on the Matterport3D dataset for comparison. We report the quantitative results in Table 3 and the qualitative results in Figure 7, which verifies the superior performance of our SpND method. This also verifies the advantages of our SpND method that arises from the spherical noise and SDC operation to accommodate panoramic deformation.

## 4.3. Ablation Study

**Effectiveness of Core Components.** We first conducted an ablation study on the Matterport3D dataset to evaluate the impact of integrating the structure prior $\mathbf{P}_{ERP}$, the spherical grid $\mathcal{R}^s$, and the learnable offsets $\Delta\mathbf{p}_n$ for the SDC layer, all of which consist of the core components in our SpND model. For simplicity, all ablation studies were conducted using the general prompt "*Outpainting*" and report the results in Table 4 and Figure 8. More specifically, we verified the effectiveness of incorporating the structural prior knowledge tailored for the ERP format, as well as the spherical

*Table 4.* Ablation study of core components in SpND model. The comparison includes three variants of our SpND model: without the structural prior (denoted as *w./o.* $\mathbf{P}_{ERP}$); without the SDC layer (denoted as *w./o.* $\mathcal{R}^s$) and the one without the learnable offsets (denoted as *w./o.* $\Delta\mathbf{p}_n$). The best results are in **bold**.

| Methods | FID ↓ | FID$_{hori}$ ↓ | Avg ↑ | Pre ↑ | Recall ↑ |
|---|---|---|---|---|---|
| *w./o.* $\mathbf{P}_{ERP}$ | 9.93 | 12.05 | 0.53 | 0.42 | 0.63 |
| *w./o.* $\mathcal{R}^s$ | 11.64 | 15.65 | 0.52 | 0.41 | 0.62 |
| *w./o.* $\Delta\mathbf{p}_n$ | 12.76 | 11.17 | 0.56 | 0.46 | 0.65 |
| SpND (ours) | **9.08** | **10.64** | **0.62** | **0.52** | **0.72** |

prior knowledge embedded in the grid $\mathcal{R}^s$ of the SDC layer. The integration of structural prior knowledge enhances the quality and diversity of completed images, as evidenced by improvements in FID, precision, and recall metrics in Table 4. The enhancements in image quality are shown in Figure 8. Furthermore, leveraging the spherical grid $\mathcal{R}^s$ within the SDC layer as the spherical prior knowledge yields additional improvements in image quality and diversity. These experimental results underscore the critical components in substantially enhancing the performance of panoramic image outpainting. Please note that, the learnable offsets, in conjunction with the spherical grid $\mathcal{R}^s$, play a crucial role in generating high-quality panoramic content, whereas the spherical prior $\mathcal{R}^s$ alone does not yield comparable results. As reported in Table 4, using the spherical grid $\mathcal{R}^s$ without the learnable offsets results in a significant degradation in both outpainting quality and diversity.

**Effectiveness of Packed Modules.** We further evaluated the effectiveness of the architecture employed in our spherical net and the prior injection layer on Matterport3D dataset. Note that without the spherical noise and the proposed SpNB, our SpND model reduces to a variant of Control-Net (Zhang et al., 2023) with the CME module (denoted as **Repaint**). Moreover, since our SDC layer uses the spherical grid similar to SphereNet (Coors et al., 2018), we further compared a variant model which replaces the deformable

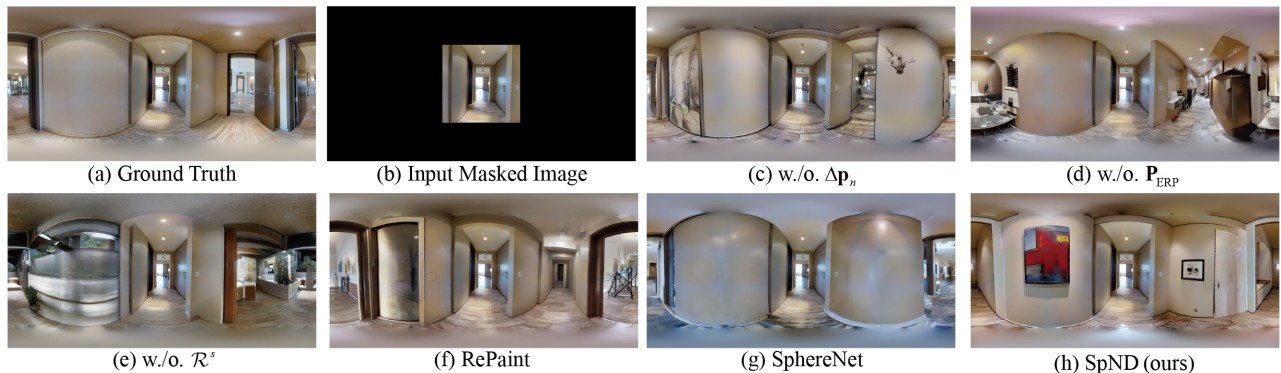

(a) Ground Truth      (b) Input Masked Image      (c) w./o. $\Delta\mathbf{p}_n$      (d) w./o. $\mathbf{P}_{\mathrm{ERP}}$

(e) w./o. $\mathcal{R}^s$      (f) RePaint      (g) SphereNet      (h) SpND (ours)

*Figure 8.* Qualitative comparison result of the ablation study on Matterport3D dataset including most our ablation settings and the ground truth. Specifically, this comparison includes 3 variants of our SpND model described in Table 4, including SpND without the structural prior (denoted as *w./o.* $\mathbf{P}_{\mathrm{ERP}}$); SpND without the SDC layer (denoted as *w./o.* $\mathcal{R}^s$) and SpND without the learnable offsets (denoted as *w./o.* $\Delta\mathbf{p}_n$), and two variants RePaint and SphereNet described in Table 5.

*Table 5.* Ablation study of packed modules in the SpND model. The comparison includes two variants: **Repaint** that utilises circular convolution without integrating the prior knowledge and **SphereNet** that replaces the deformable convolution and the SDC layer with spherical convolution proposed in (Coors et al., 2018). The best results are in **bold**.

| Methods | FID ↓ | FID$_{\mathrm{hori}}$ ↓ | Avg ↑ | Pre ↑ | Recall ↑ |
|---|---|---|---|---|---|
| Repaint | 10.10 | 12.89 | 0.58 | 0.42 | **0.74** |
| SphereNet | 11.78 | 15.74 | 0.54 | 0.41 | 0.67 |
| SpND (ours) | **9.08** | **10.64** | **0.62** | **0.52** | 0.72 |

*Table 6.* Ablation study evaluating the impact of varying sampling densities $\zeta$ quantitatively on the Matterport3D dataset. The best results are in **bold**.

| Sampling Density | FID ↓ | FID$_{\mathrm{hori}}$ ↓ | Avg ↑ | Pre ↑ | Recall ↑ |
|---|---|---|---|---|---|
| $\zeta = 15$ | 12.74 | 15.52 | 0.52 | 0.38 | 0.65 |
| $\zeta = 30$ | 9.08 | 10.64 | **0.62** | **0.52** | **0.72** |
| $\zeta = 45$ | **8.84** | **10.27** | **0.62** | 0.51 | **0.72** |
| $\zeta = \infty$ (*w./o.* $\mathbf{P}_{\mathrm{ERP}}$) | 9.93 | 12.05 | 0.53 | 0.42 | 0.63 |

convolution and the SDC layer in the SpNB with the spherical convolution proposed in (Coors et al., 2018) (denoted as **SphereNet**). We report the results in Table 5 and Figure 8. From the table and figure, the proposed SpND model, which integrates the sphere nature in a *micro way*, significantly enhances the quality of the generated content. However, this improvement comes at the cost of a slight reduction in outpainting diversity. Overall, the outpainting performance is notably improved, as evidenced by the FID metric in Table 5. More importantly, our model consistently achieves the best performance, demonstrating that directly employing spherical convolution fails to effectively integrate with the pre-trained diffusion model.

**Quantitative Evaluations for Spherical Noise.** The non-*i.i.d.* noise, which serves as explicit structural prior knowledge tailored for the ERP format, constitutes a key com-

ponent of our SpND model, with the parameter $\zeta$ playing a pivotal role as mentioned in Section 3.1. To conduct a quantitative analysis of how sampling density influences the performance, we further performed an ablation study by varying $\zeta$ and report the results in Table 6. Here, $\zeta = \infty$ corresponds to the *w./o.* $\mathbf{P}_{\mathrm{ERP}}$ configuration in Table 4, which employs the widely applied *i.i.d.* noise. As shown in Table 4, our SpND method achieves optimal performance at $\zeta = 45$ and $\zeta = 30$, consistent with the analysis in Appendix A. In particular, $\zeta = 30$ achieves performance comparable to $\zeta = 45$ while offering improved computational efficiency. A reduced sampling density ($\zeta = 15$) marginally degrades performance but still surpasses existing baseline methods. These results confirm the rationale and robustness of our proposed spherical noise.

## 5. Conclusion

In this paper, we have proposed the novel spherical-nested diffusion (SpND) model, that for the first time integrated panoramic characteristics within the design of network architecture, namely, in a *micro way*. More specifically, we first developed the spherical noise catering for the structural prior of panoramic images. To capture correct content within panoramic format, we have adopted the spherical grid within deformable convolution and have proposed the spherical deformable convolution (SDC) layers. Leveraging the structural prior and the SDC layer, we have introduced the SpND model which integrated the spherical net with a pre-trained diffusion model to achieve superior outpainting for panoramic images. Extensive experiments have verified the superior performances for outpainting high-quality and diverse panoramic images, surpassing all the baselines. We may also need to point out our limitations by using Repaint technique in the latent space, which leads to slight distortion between the input and output against masks. Future works are expected to further refine the mask areas during the diffusion process.

## Acknowledgement

This work was supported by Natural Science Foundation of China (NSFC) under Grants 62450131, 62206011, 62231002 and 62401027, and Beijing Natural Science Foundation under Grant L223021.

## Impact Statement

This paper presents work whose goal is to advance the field of Machine Learning. There are many potential societal consequences of our work, none which we feel must be specifically highlighted here.

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

# A. Analysis on the Structural Prior for the ERP Format

Compared to the equatorial region, rows closer to the poles correspond to a smaller number of points on the sphere being mapped onto the fixed $m_w$ points in the ERP format. This discrepancy results in a one-to-many mapping, where multiple pixels in the ERP representation correspond to the same point in the 3D space, leading to redundancy and distortion in these regions. Since the number of points per row ($m_w$) in ERP format is given by the width $w$, further analysis on (2) reveals that as the resolution of the ERP format decreases or the sampling density $\zeta$ in 3D space increases, the distribution of the uniform spherical noise more closely approximates *i.i.d.* Gaussian noise within the ERP representation.

To comprehensively analyse the properties and the influence of the hyperparameter $\zeta$ and the width $w$ on the structural prior, we conducted an experiment and present the results in Figure A.1. Note that images with a resolution of $128 \times 64$ are consistent with the resolution of latent features in our SpND model, while images with a resolution of $1024 \times 512$ match the resolution of the images in datasets. By comparing the columns in Figure A.1, it can be concluded that structural prior with higher resolution produced using the same $\zeta$ in the ERP format exhibit fewer rows that satisfy the *i.i.d.* property, consistent with our analysis. Additionally, by comparing rows with the same resolution in Figure A.1, it can be observed that noise in 3D space with smaller $\zeta$ tends to produce a structural prior with fewer rows satisfying the *i.i.d.* property at the same resolution. For $\zeta = 1000$ and a resolution of $128 \times 64$, the structural prior closely approximates *i.i.d.* Gaussian noise, with only the top and bottom rows significantly violating the *i.i.d.* property. Since the noise is adopted in the latent space in our SpND model, we adopt the noise distribution in the ERP format with a resolution of $128 \times 64$ as our structural prior.

To determine the optimal value of the hyperparameter $\zeta$, we conducted a comparison between the visualised noise and the latent features, as illustrated in Figure A.2. This analysis led us to select $\zeta = 30$, as it most accurately preserves the structural characteristics of the feature map with a resolution of $128 \times 64$ while maintaining the highest efficiency in panorama outpainting.

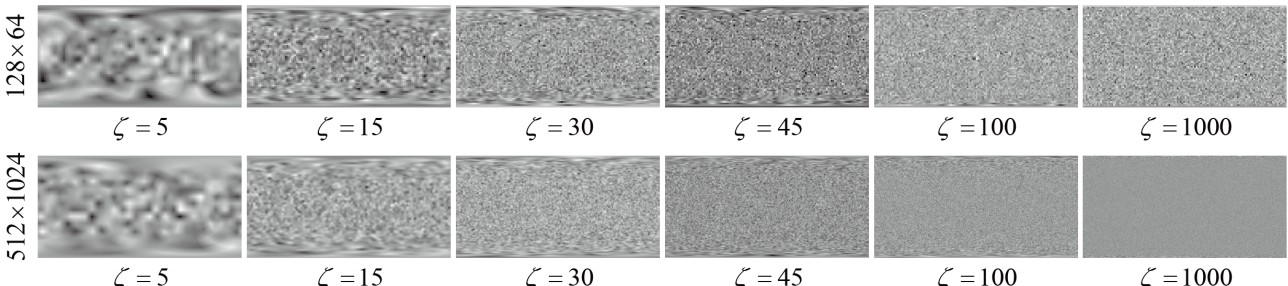

*Figure A.1.* Analysis on the impact of resolution (width $w$, height $h$) and sampling density $\zeta$ in 3D space to the structural prior for the ERP format. The structural prior is obtained by projecting the *i.i.d.* Gaussian noise in 3D space with varying density $\zeta$, into the ERP format. The first row shows projections with a resolution of $128 \times 64$, while the second row presents those with a resolution of $1024 \times 512$.

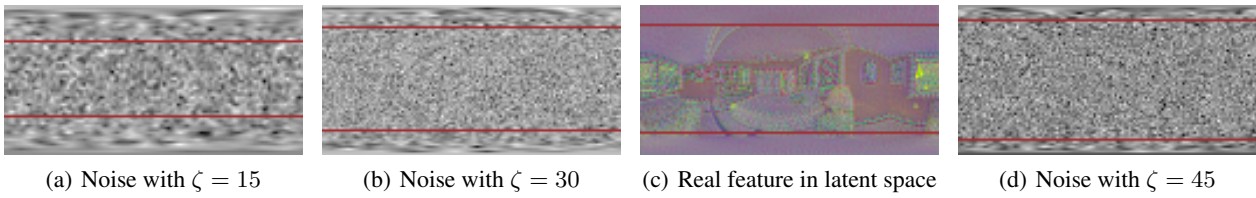

    (a) Noise with $\zeta = 15$      (b) Noise with $\zeta = 30$      (c) Real feature in latent space      (d) Noise with $\zeta = 45$

*Figure A.2.* Comparison of noise with different values of $\zeta$ and the real feature in the latent space encoded by the VAE. The **red line** is used to distinguish rows that approximate the *i.i.d.* property from those that do not, thereby highlighting regions where the noise significantly deviates from the characteristics of *i.i.d.* Gaussian noise.

Since a specially designed noise pattern was employed as structural prior knowledge for the ERP format, this study aimed to analyse its impact on the resulting outputs. As demonstrated in (a) and (c) of Figure A.3, changing the noise used to realize the structural prior $\mathbf{P}_{\mathrm{ERP}}$ does not affect the generated content. In contrast, changing the noise $\mathbf{Z}$ used for the pretrained diffusion model primarily impacts the generated content, as illustrated in (b) and (d) in Figure. A.3. This indicates that the noise pattern associated with $\mathbf{P}_{\mathrm{ERP}}$ primarily captures the spherical curvature, instead of controlling the specific details of the generated content. Note that the masked input is identical in (a) and (b), as well as in (c) and (d) in Figure A.3.

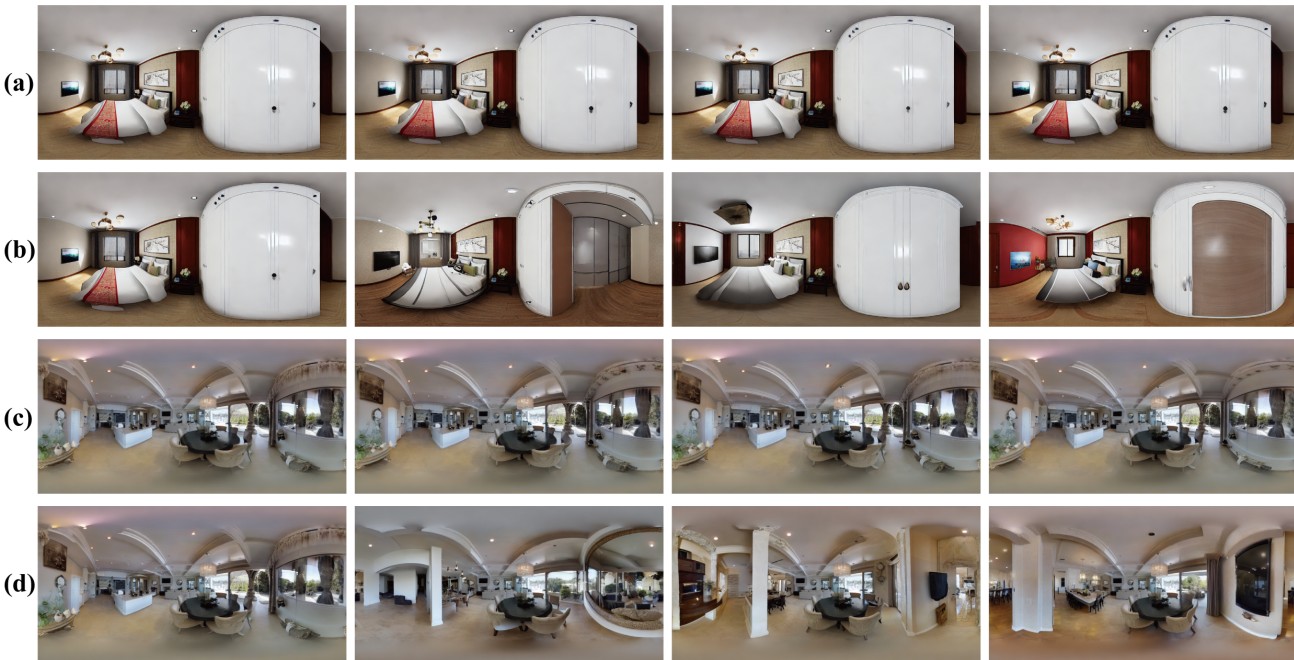

*Figure A.3.* The impact of the structural prior $\mathbf{P}_{\text{ERP}}$ on the outputs is examined by sampling different noise patterns in 3D space as the structural prior $\mathbf{P}_{\text{ERP}}$, while keeping the feature $\mathbf{Z}$ consistent across rows (a) and (c). In rows (b) and (d), the same 3D noise is used for the structural prior, but the feature $\mathbf{Z}$ is varied. Rows (a) and (b) present samples from the Structured3D dataset, while rows (c) and (d) display samples from the Matterport3D dataset.

## B. Repaint Operation with Mask Control

The repaint operation (Lugmayr et al., 2022) involves adopting a mask $\mathbf{m}$ to regulate specific regions within the generated sample while preserving the remaining areas unchanged. Specifically, the mask $\mathbf{m}$ determines the regions that should remain unchanged by assigning a value of 1, while the remaining areas, represented by a value of 0, are updated during the diffusion process. At each timestep $t$, our SpND model predicts the denoised sample $\mathbf{Z}_{t-1}^{\text{predicted}}$, which can be formulated as

$$\hat{\mathbf{Z}}_{t-1}^{\text{predicted}} = \frac{1}{\sqrt{\alpha_t}}\left[\mathbf{Z}_t^{\text{predicted}} - \frac{1-\alpha_t}{\sqrt{1-\bar{\alpha}_t}}\epsilon_\psi\left[\mathbf{Z}_t^{\text{predicted}}, t\right]\right],\tag{B.1}$$

where $\bar{\alpha}_t = \prod_{i=1}^{t}\alpha_i$ and The parameter $\alpha$ serves as a hyperparameter that controls the variance of the noise. Assume that $\eta$ denotes for the *i.i.d.* Gaussian noise and $\mathbf{Z}_0$ corresponds to the ground-truth feature. Subsequently, to achieve mask control, we update the denoised sample $\hat{\mathbf{Z}}_{t-1}^{\text{predicted}}$ given $\mathbf{Z}_{t-t}^{\text{real}}$ according to the formula as

$$\mathbf{Z}_{t-1}^{\text{predicted}} = \mathbf{m} \odot \hat{\mathbf{Z}}_{t-1}^{\text{predicted}} + (1-\mathbf{m}) \odot \mathbf{Z}_{t-t}^{\text{real}},\tag{B.2}$$

$$\mathbf{Z}_{t-t}^{\text{real}} = \sqrt{\bar{\alpha}_{t-1}} \cdot \mathbf{Z}_0 + \sqrt{1-\bar{\alpha}_{t-1}} \cdot \eta\tag{B.3}$$

where $\mathbf{Z}_{t-1}^{\text{predicted}}$ represents the outpainted feature and $\odot$ denotes element-wise multiplication. Note that (B.2) formulates the repaint process and (B.3) formulates the forward process of the diffusion model. The repaint operation, as defined in (B.1)-(B.3), ensures that the masked regions retain their original values, while the unmasked regions are selectively updated. Note that this effect of the repaint operation aligns seamlessly with the objective of the image outpainting task. By leveraging the repaint technique, our SpND model effectively performs outpainting for panoramic images while preserving the integrity of the original part of the image.

## C. Experiment on Martain Image Outpainting

We further conducted experiments on a small dataset consisting of Martian panoramic images to evaluate the generalizability and the robustness of our SpND model. Specifically, we collected 145 Martian panoramic images captured by the

Perseverance rover, with a resolution of $512 \times 172$, from publicly available sources. These images were padded with black borders to a final size of $512 \times 256$. As shown in Figure C.1, the proposed $SpND_{prompt}$ model effectively addresses the black borders at the top and bottom of the image, generating high-quality panoramic content for Martian images with a limited number of training samples. Consistent with our other experiments, we employed the BLIP-2 model (Li et al., 2023) to generate the prompts for the Martian images. Note that the masked input can be obtained by projecting view images into the ERP format.

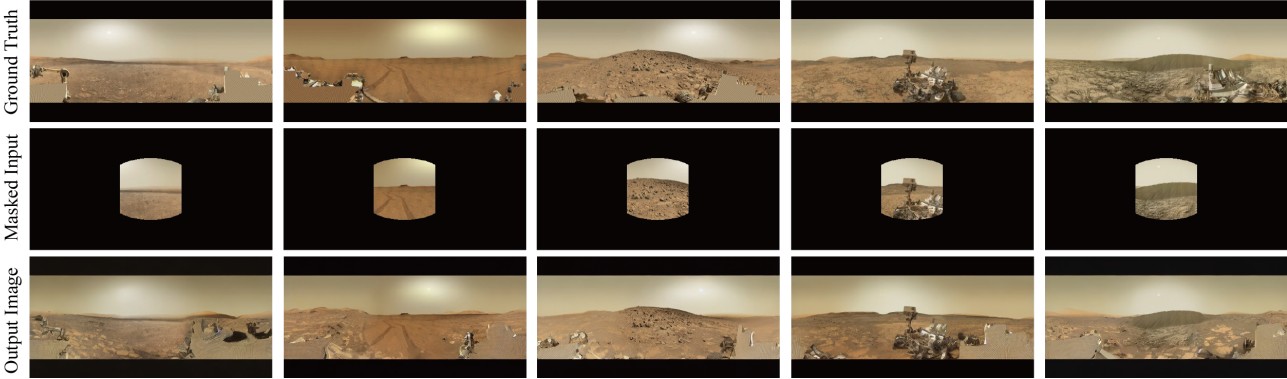

*Figure C.1.* Qualitative results of panoramic image outpainting on the Martian dataset based on the given perspective view.

## D. Experiment on Input Versatility

In addition, we have demonstrated that our model can accommodate multiple partial-view and overlapping inputs by adjusting the training masks accordingly on the Matterport3D dataset. More specifically, for the multi-input scenario, we evaluated our SpND method based on two viewpoints centred at $(\theta_1 = -90°, \phi_1 = -15°)$ and $(\theta_2 = 90°, \phi_2 = 15°)$ within the ERP format, thus denoted as **Dual**. Moreover, for the overlapping scenario, we evaluated our SpND method on two viewpoints centered at $(\theta_1 = 20°, \phi_1 = 0°)$ and $(\theta_2 = 90°, \phi_2 = 15°)$ with overlapped regions, also using the ERP format, thus denoted as **Overlapping**. We then report the subjective results in Figure D.1 and Figure D.2. As illustrated in the figures, the proposed $SpND_{prompt}$ model is capable of achieving high-quality panoramic image completion under varying masks and multiple input conditions. We can conclude that our method can generalize to multiple overlapping and partial-view inputs.

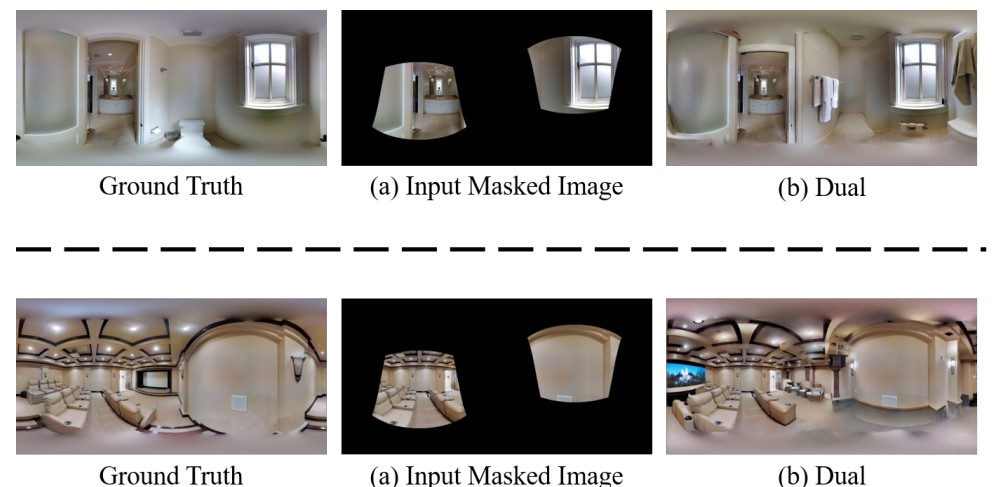

*Figure D.1.* Qualitative results of $SpND_{prompt}$ with multiple partial-view inputs, including ground truth, masked input, and images completed by our $SpND_{prompt}$.

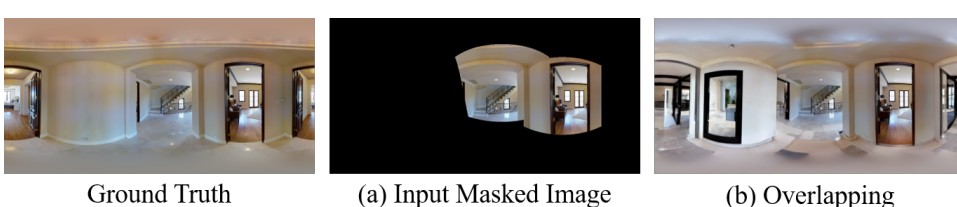

|  |  |  |
|:---:|:---:|:---:|
| Ground Truth | (a) Input Masked Image | (b) Overlapping |

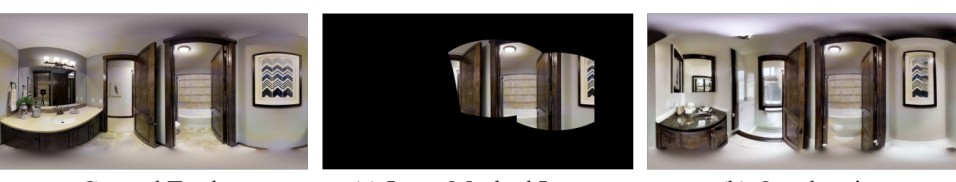

|  |  |  |
|:---:|:---:|:---:|
| Ground Truth | (a) Input Masked Image | (b) Overlapping |

*Figure D.2.* Qualitative results of SpND$_{prompt}$ with multiple overlapping inputs, including ground truth images, masked input images, and images completed by our SpND$_{prompt}$.

