# OpenReview forum: "Spherical-Nested Diffusion Model for Panoramic Image Outpainting"
_ICML.cc/2025/Conference — ICML 2025 poster_

### Official Review · Reviewer_RXh3 · 2025-03-09

**Overall Recommendation:** 3

**Summary:**

This paper presents a new diffusion-based panoramic image outpainting control net model that incorporates  1) spherical noise as structural prior into the equirectangular projected image for better handling the ERP distortion and 2) spherical deformable convolution layers to handle the varying CNN reception field on the ERP image. The experimental results demonstrate the promising panorama outpainting quality compared to prior works.

**Claims And Evidence:**

The claims made in the paper are supported by clear and convincing evidence.

**Essential References Not Discussed:**

References are sufficient.

**Experimental Designs Or Analyses:**

The experimental designs and analyses are sound to me.

**Methods And Evaluation Criteria:**

Yes, this submission largely follows the evaluations from the prior work.

**Other Comments Or Suggestions:**

None

**Other Strengths And Weaknesses:**

Strengths:
* The introduced spherical noise as a structural prior for panorama diffusion denoising is interesting and novel.
* The design of spherical deformable convolution is reasonable and effective.

Weaknesses:
* Since the outpainting tasks largely rely on perceptual evaluations, the author should provide more visual results to assist readers’ perceptual comparison. Besides, there’s no visual result in the ablation study. Providing the reference ground truth would also be helpful.
* Since the evaluated mask region is located at the center of the ERP panorama, input image patches do not contain too much distortion. This evaluation setting may not show the advantage of spherical noise and SDC processed on meaningful pixels.

**Questions For Authors:**

* How are the panorama images masked during training? Are they only masked around the center? What about the range of elevation angle?
* How does this model generalize to varying masks? If the entire input is masked, will this model work properly?
* It would be better if the authors could include more applications of the proposed method.

**Relation To Broader Scientific Literature:**

The proposed method of injecting spherical structural noise to the original diffusion noise might be relevant to the non-isotropic diffusion model, for example:
* “Score-based generative modeling with critically-damped langevin diffusion”
* “Score-based denoising diffusion with non-isotropic gaussian noise models”

**Theoretical Claims:**

There’s no proof for theoretical claims.

---

> ### Author Rebuttal · Authors · 2025-04-01
>
> Many thanks for your positive opinion and valuable comments!
> - **Tab.1 - 4 are in https://anonymous.4open.science/api/repo/tabl/file/T.pdf?v=5327f2ef**
> - **Fig.1 - 9 are in https://anonymous.4open.science/api/repo/figu/file/F.pdf?v=2d1be253**
>
> > ### **1 Supplementary Material: Further Analysis on Fig. A. 3**
>
> The randomness in our SpND model consists of two components: the structural prior $\mathbf{P}\_\text{ERP}$ from spherical noise and the diffusion noise $\mathbf{Z}$. Fig. A. 3 in Supplementary-A illustrates the impact of these two components, in which spherical noise minimally affects the semantic content of generated images, maintaining structural consistency. In contrast, diffusion noise drastically changes the output, creating distinct visual features. This indicates spherical noise serves as a structural prior, preserving geometric constraints, while diffusion noise controls semantic content generation. These observations support our formulation of $\mathbf{P}\_\text{ERP}$ as a mechanism to regularize the latent space without disturbing semantic attributes.
> > ### **2 Relation to Broader Scientific Literature: Relation to Non-Isotropic Diffusion Models**
>
> The noise in our SpND method follows non-isotropic Gaussian distributions, with the non-isotropic property arising from the ERP operation applied to spherical noise. In contrast, Ref. [1] introduces a critically-damped Langevin diffusion process to enhance network convergence by adding an auxiliary velocity variable, while Ref. [2] generalizes score-based denoising diffusion models with non-isotropic Gaussian Free Field (GFF) noise, improving generation quality on CIFAR10. However, these methods, with their particular non-isotropic settings, are not designed for panoramic image outpainting. For instance, GFF noise in Ref. [2] defines the covariance between sampled values $f(\theta, \phi)$ and $f(\theta', \phi')$ as:  $$\left(\text{Cov}(f(\theta,\phi), f(\theta',\phi')) \approx -\log\|(\theta,\phi)-(\theta',\phi')\| \right) + C,$$where $C$ is a constant. In contrast, for our spherical noise, when $\phi=\frac{\pi}{2}$, the covariance simplifies to: $$\text{Cov}(f(\theta,\frac{\pi}{2}), f(\theta',\frac{\pi}{2})) = \sigma^2.$$Thus, the constant covariance $\sigma^2$ contradicts the logarithmic dependence required by Ref. [2], demonstrating that the non-isotropic noise in our SpND method cannot be generalized by existing models. This highlights the novelty and contribution of our approach in panoramic outpainting.
> > ### **3 Weakness 1: More Visual Results**
>
> Yes, outpainting tasks heavily rely on perceptual evaluations. We have added more visual examples for all comparisons discussed in our experiments, including the main comparisons, ablation study and view-image outpainting experiments, in Figs. 5-8 in the link. The results contain ground truth, masked input, all baseline methods, and our SpND method for both the Matterport3D and Structured3D datasets, which consistently demonstrate the superior performance of our SpND method.
> > ### **4 Weakness 2: Evaluations on Distorted Masks**
>
> To further evaluate our method on different masks, we conducted additional experiments on masks near the poles, where spherical distortion becomes evident. Specifically, we projected a $256 \times 256$ image onto the ERP format at $\theta=90^\circ, \phi=60^\circ$ to create a highly-distorted mask near the pole. We then retrained our SpND method, along with the second and third best baseline methods (PanoDiff and PanoDiffusion). Results reported in Tab. 4 and Fig. 9, confirm the superior performance of our SpND method, highlighting the advantages of spherical noise and the SDC operation in accommodating panoramic deformation.
> > ### **5 Questions For Authors 1: Mask Location and Evaluation Angle**
>
> Yes, in our original experiments, the images were masked only around the center during training, corresponding to Fig. 4-(a) in our manuscript. For the masked input, the region containing image content spans an elevation angle range of approximately $[-45^\circ, 45^\circ]$, within a full image that has an elevation angle range of $[-90^\circ, 90^\circ]$. We further evaluated our SpND method's ability to generalize to various non-centered masks, as detailed in **Rebuttal 6 below**.
> > ### **6 Questions For Authors 2: Generalization to Varying Masks**
>
> Yes, our SpND model can generalize to varying mask formats. We re-trained our SpND method with the entire input masked and show the subjective results in Fig. 4. While performance was reduced, our SpND method achieved an FID of 29.65 and successfully performed panoramic generation, confirming its generalization ability. Additionally, our SpND method is highly flexible with multiple input views and we refer to our **Rebuttal 4 above** and our **Rebuttal 3 to Reviewer Qs7j**.
> > ### **7 Questions For Authors 3: More Possible Applications**
>
> Many thanks! For more possible applications, we refer to our **Rebuttal 1 to Reviewer GNkK**.

---

> > ### Comment · Reviewer_RXh3 · 2025-04-08
> >
> > Thanks for the authors' response and additional results. Most of my concerns are addressed in the rebuttal. I appreciate the efforts in preparing the additional results on varying non-trivial out-painting. At least compared to the baselines, these results are competitive. I will keep my positive rating on this submission.

---

> > > ### Author Response · Authors · 2025-04-08
> > >
> > > We wish to thank the reviewer very much for taking the time to read our responses and provide valuable comments! And we are also glad that we have addressed your concerns.

---

### Official Review · Reviewer_vstk · 2025-03-13

**Overall Recommendation:** 4

**Summary:**

This work proposes SpND, a novel pipeline for panoramic image outpainting based on diffusion model. The authors tackle the limitation of previous works in which the spherical nature of panoramic image is injected using soft regularization techniques, often failing to fully enforce the spherical constraint. To address this, this work aims to intrisically enforce the sphercial constraint during the generation through a sphercial-nested diffusion model, including the sphereical noise technique and a novel spherical deformable convolution (SDC) specifically designed to handle the spherical grid of panoramic images. SpND shows the state-of-the-art performance on panoramic image outpainting, surpassing previous works on image quality.

**Claims And Evidence:**

1. The paper is well written and easy follow. The authors tackle a practically meaningful application of panoramic outpainting.

2. The two main technical contributions - spherical noise and SDC - both have clear motivations and seem to be effective based on the qualitative/quantitative comparisons and ablations studies.

3. The analysis on the spherical property of the ERP format well delivers why it is critical to sample the noise in the spherical spaces instead of directly from the planar space.

4. Yet, it would be better if the authors provide a clear elaboration on the claim that "one-to-many mapping inherently introduces spatial coherence in the EPR domain". While these overlaps between multiple views can enforce the same content to be generated in the corresponding pixels, it is unclear how this can also lead to a globally coherent image. Is it because the coherent content/style is somehow propagated across different views during denoising? Moreover, have the authors observed any downsides of this one-to-many mapping, such as blurry/over-smoothed image near the overlapping regions?

**Essential References Not Discussed:**

As mentioned above, an important related work PanFusion [1] is missing. Since this is not an outpainting method but rather a pure text-based generation method, it doesn't seem necessary to make quantitative comparions. But it would be nice to discuss why this previous approach is not sufficient for panoramic outpainting.

Moreover, AOG-Net [2] seems to be a more directly related work, and possibly a valid baseline.

[1] Taming Stable Diffusion for Text to 360◦ Panorama Image Generation, Zhang et al., CVPR 2024

[2] Autoregressive Omni-Aware Outpainting for Open-Vocabulary 360-Degree Image Generation, Lu et al., AAAI 2024

**Experimental Designs Or Analyses:**

1. The paper provides detailed explanations on both the evaluation settings and implementation details of SpND.

2. Both quanitative and qualitative comparisons are presented clearly, showing that SpND outperforms previous methods on panoramic outpainting.

3. I am confused by the choice of using FID and FID_hori for image quality evaluation. To measure the quality and realism of the panoramic image, it seems more intuitive to either measure the FID of the projected perspective views or use a panorama-specific metric such as FAED [1].

4. The paper provides a thorough ablation study of its core components, which shows that the design choice for the pipeline works according to their intentions.

[1] Bips: Bi-modal indoor panorama synthesis via residual depth-aided adversarial learning, Oh et al., ECCV 2022

**Methods And Evaluation Criteria:**

1. The introduction of SDC seems to be quite novel and an approach for handling the spherical nature of panoramic images. Adopting the idea of deformable convolution specifically for the spherical grid seems to be a reasonable method that takes into account the key characteristic of panoramic images.

2. While this work aims for panoramic "outpainting", the design choice of SpND would be better justified if the authors had provided discussions that compare with the panoramic generation works. For instance, MVDiffusion [1] and PanFusion [2] both suggest novel architectures for text-to-panorama generation. Can we also apply RePaint to such models to obtain panoramic image outpainting? If it is not possible to apply the same approach to these works, can the authors please explain which aspect of SpND allows this in contrast?

3. Moreover, based on the pretrained SpND model, the RePaint technique is utilized to perform seamless outpainting based on the input mask. Since the original RePaint is based on "pixel-space" diffusion models, it doesn't introduce any discrepancy between the input mask and the space in which the denoising is performed. In constrast, since this work uses a latent diffusion model, I am concerned if the discrepancy between the input mask (pixel-space) and the actual inpainting operations (latent-space) could lead to quality degradation. If this is not the case, it would be nice if the authors can explain why this discrepancy does not necessarily lead to degradation.

[1] MVDiffusion: Enabling Holistic Multi-view Image Generation with Correspondence-Aware Diffusion, Tang et al., NeurIPS 2023

[2] Taming Stable Diffusion for Text to 360◦ Panorama Image Generation, Zhang et al., CVPR 2024

**Other Comments Or Suggestions:**

When first reading the paper, it was a bit confusing whether the whole pipeline is trained end-to-end and the inference is done in a simple feed-forward manner. However, as far as I understand, the inference for outpainting is done by applying iterative RePaint steps using the pretrained diffusion model. It would have been better if that was emphasized.

**Other Strengths And Weaknesses:**

The strengths and weaknesses are discussed in the above sections.

**Questions For Authors:**

Some questions are already included in the above sections.

**Relation To Broader Scientific Literature:**

It would be interesting to see whether the proposed idea of inject a structural prior by sampling from spherical noise could be extended to pure generation of panoramic images. For instance, can it lead to more accurate panoramic images (or videos) generated just from text prompts? Applying a similar idea to text-to-panorama methods like MVDiffusion [1] and PanFusion [2] would be an interesting future work.

[1] MVDiffusion: Enabling Holistic Multi-view Image Generation with Correspondence-Aware Diffusion, Tang et al., NeurIPS 2023

[2] Taming Stable Diffusion for Text to 360◦ Panorama Image Generation, Zhang et al., CVPR 2024

**Theoretical Claims:**

This work does not provide theoretical claims and instead focuses on empirical evidence for the effectiveness of SpND.

---

> ### Author Rebuttal · Authors · 2025-04-01
>
> Many thanks for the positive comments and insightful suggestions.
> - **Tab.1 - 4 are in https://anonymous.4open.science/api/repo/tabl/file/T.pdf?v=5327f2ef**
> - **Fig.1 - 9 are in https://anonymous.4open.science/api/repo/figu/file/F.pdf?v=2d1be253**
>
> > ### **1 Claims and Evidence 4: Elaboration on One-to-Many Mapping**
>
> The one-to-many mapping naturally arises from the ERP format of the spherical content. Pixels near the polar regions exhibit strong correlations in the ERP domain, reflecting spatial coherence in this projection. The consistent denoising procedure ensures coherent style and content by propagating the entire image across different views. Controlled by $\zeta$, the one-to-many correspondence reaches its optimum when the correlations between noise and feature/image are similar, as explained in Supplementary-A. When $\zeta$ is too small, blurry artifacts occur due to excessive pixel correspondence. Conversely, when $\zeta$ is too large, spatial coherence decreases. This was verified through two new ablation studies, which is detailed in **Rebuttal 2 to Reviewer QS7j**.
>
> > ### **2 Methods and Evaluation Criteria 2: Applicability of RePaint to Panoramic Generation Methods**
>
> Outpainting can be achieved through text-to-panorama and diffusion-based generative methods, with each inference step integrated by the RePaint operation for denoising the full panoramic image. However, MVDiffusion generates $8$ perspective views via a multi-view diffusion model, which are then aggregated into a full panorama using Rodrigues rotation. This makes it infeasible to access each inference step for denoising, preventing the straightforward application of RePaint. In contrast, PanFusion accesses the full latent panoramic features at each inference step, enabling outpainting via RePaint. We modified PanFusion by adding RePaint and present new results in Fig. 3 in the link. As shown, PanFusion's outpainted panoramas exhibit repetitive patterns and poor quality due to latent feature rotation to preserve spherical continuity. In comparison, our SpND uses a masked image as input and consistently achieves the superior panoramic outpainting.
>
> > ### **3 Methods and Evaluation Criteria 3: Pixel-space Discrepancy by RePaint**
>
> Yes. The discrepancy may exist in our SpND method. Most diffusion-based outpainting methods, particularly for panoramic images, operate within the latent space and suffer accuracy degradation in the input mask. For methods using RePaint, we calculated the PSNR$\_\text{cent}$ for the repainted area against the original image, with results reported in Tab. 3 in link. As shown, our SpND method achieves the highest accuracy and quality, which naturally results from our intrinsic SDC operation and spherical noise.
>
> > ### **4 Experimental Designs or Analyses 3: Clarifying FID and FID$_\text{hori}$, with Enhanced Metrics**
>
> The FID metric is widely used to evaluate generated image quality, and we preliminarily applied it to assess the performance of outpainted panoramic images. The FID$_\text{hori}$ metric evaluates the FID values of 8 perspective view images of size $512 \times 512$, following MVDiffusion. We also incorporated the panorama-specific FAED metric to evaluate the quality of panoramic images. Notably, FAED requires depth information, which is infeasible for outpainting. Therefore, we followed the PanFusion method to compute FAED without explicit depth input. The results, reported in Tab. 3, demonstrate that our SpND method outperforms state-of-the-art outpainting techniques.
>
> > ### **5 Relation to Broader Scientific Literature: Applicability to Panoramic Generation Tasks**
>
> Yes, we believe injecting non-*i.i.d.* noise into existing panoramic generation methods could further improve detail retention and the quality of generated panoramic images, including text-to-image methods such as  MVDiffusion and PanFusion. Our SpND method can also generalize to generate images from text descriptions using a fully masked input. We evaluated this during the rebuttal and we achieved an FID score of 29.65 by retraining our method without other modifications. Subjective results are shown in Fig. 4.
>
>
> > ### **6 Essential References not Discussed: PanFusion and AOG-Net**
>
> Yes, we have analyzed PanFusion by using the RePaint strategy, by **Rebuttal 2 above**. AOG-Net uses an autoregressive pipeline for 360° outpainting, utilizing feature remapping for panoramic content. We conducted a comparative analysis in Fig. 5 and Tab. 3 against AOG-Net using the Matterport3D dataset. From these, we conclude that our SpND method, by enforcing panoramic deformation through intrinsic convolution and spherical noise, consistently outperforms others in generating high-quality outpainted panoramic images.
>
> > ### **7 Other Comments or Suggestions: Training and Inference Procedures**
>
> Yes. During inference, we applied iterative RePaint steps based on the trained SpND model, following other outpainting methods such as PanoDiffusion.

---

> > ### Comment · Reviewer_vstk · 2025-04-05
> >
> > I appreciate the authors for thoroughly addressing the raised concerns, particularly by providing additional experiment results on PanFusion and AOG-Net.
> >
> > My concerns regarding the advantage of SpND over previous diffusion-based methods have been resolved, and I have updated my recommendation to "Accept".

---

> > > ### Author Response · Authors · 2025-04-05
> > >
> > > Thank you very much for providing valuable comments and reading our responses. We are glad that we've addressed your questions! In our revised paper, we will further improve our experimental settings and evaluations against the state-of-the-art baselines, to further verify the advantage of our SpND against existing diffusion-based methods.

---

### Official Review · Reviewer_GNkK · 2025-03-15

**Overall Recommendation:** 3

**Summary:**

This paper proposes a spherical reformulation of diffusion models for panoramic image outpainting. It focuses on the fact that the processing unit and the spatial stochastic patterns in plain diffusion models do not align well with panoramic image outpainting. To handle this problem, the paper proposes redesigning some components, such as spherical noise, spherical deformable convolution, and circular mask. This design choice is different from existing works, which utilize soft constraints. Experimental results clearly show that the proposed method outperforms the existing methods.

## update after rebuttal

All the reviewers have acknowledged the contribution of the paper. Although I had some doubts regarding the broader impact, I also think that this is a valuable reformulation for an important use case. I maintain my original score.

**Claims And Evidence:**

The claim that panoramic outpainting requires components that can naturally deal with spherical patterns, is plausible and convincing. Using tailored modules rather than soft constraints can obviously give better results.

**Essential References Not Discussed:**

I believe the bibliography is thorough enough.

**Experimental Designs Or Analyses:**

As mentioned above, experimental designs are sound.  Experimental results show that the proposed method outperforms existing methods, supporting the effectiveness of the proposed modules.

**Methods And Evaluation Criteria:**

The proposed modules or changes (e.g., spherical noise, spherical deformable convolution, and circular mask) are quite intuitive and well-designed. The paper provides most of the standard evaluation metrics for popular benchmark datasets.

**Other Comments Or Suggestions:**

There are some typos, e.g., "gird" -> "grid."

**Other Strengths And Weaknesses:**

As I mentioned above, this is a good paper focusing on a specific problem. The proposed method is well designed and the performance is great, so I have no doubt in this part. I think the main question boils down to its broader impact. This method is somewhat limited to this particular problem, and the novelties in the method are not like "groundbreaking." But at the same time, the problem itself has many valuable real-life applications. At the moment, I'm slightly leaning towards the positive side.

**Questions For Authors:**

(9) is somewhat puzzling. Seeing Fig. 3, it seems the outputs of Spherical Net are "embedded" in the main U-Net. This seems slightly contradictory to (9). Could the authors clarify this point?

**Relation To Broader Scientific Literature:**

I believe this paper proposes a well-designed method tailored to a particular problem (panoramic outpainting). For broader fields, the impact is somewhat limited. That being said, the problem being dealt with in this paper has some practical values for various applications (VR, AR, etc.). Overall, my opinion is that this is a good paper focusing on a specific problem.

**Theoretical Claims:**

N/A

---

> ### Author Rebuttal · Authors · 2025-04-01
>
> We wish to thank the reviewer very much for the positive opinion and insightful suggestions.
> - **Tab.1 - 4 are in https://anonymous.4open.science/api/repo/tabl/file/T.pdf?v=5327f2ef**
> - **Fig.1 - 9 are in https://anonymous.4open.science/api/repo/figu/file/F.pdf?v=2d1be253**
>
> > ### **1 Weaknesses: Our Broader Impact.**
>
> Yes. Although belonging to the broader fields of panoramic image processing, the panoramic outpainting task still possesses rich applications, especially for nowadays VR/AR applications such as autonomous driving, medical imaging, cultural heritage preservation, media and entertainments, to name but a few. The capability of completing panoramic content from masked views also allows for efficient compression and transmission of large-volume panoramic images, one of the key requirements to achieve meta-universe. Moreover, the newly proposed techniques, such as the spherical deformable convolution (SDC) operation and spherical noise, may also contribute to other panoramic-related tasks, including panoramic generation (as we have newly evaluated in **Rebuttal 5 to Reviewer vstk**) and content understanding. We believe that this direction is worth investigating with rich juicy unrevealed, in which our method paves a possible way to the field.
>
> > ### **2 Other Comments Or Suggestions: Typos**
>
> Many thanks! We will modify ''gird'' to ''grid'' and also carefully polish our wordings, by correcting grammatical errors and typos.
>
> > ### **3 Questions For Authors: Clarification on Equation (9)**
>
> Many thanks for the careful considerations. Yes, the outputs of our spherical net are integrated into the primary U-Net architecture, as depicted in Fig. 3 of our manuscript. Therefore, Equation (9) was misleading and is corrected by:
> $$
> \epsilon\_{\boldsymbol{\psi}} = f\_\text{SD}\left[\mathbf{Z}\_{t}, \mathbf{T}, f\_\text{PDN}\left[\mathbf{F}\_\text{pri}, \mathbf{T}\right]\right]
> $$
> where $f\_{\text{SD}}$ denotes the pre-trained diffusion model and $f\_{\text{PDN}}$ denotes the spherical net, which is embedded into the diffusion model. Moreover, $\mathbf{T}$ is the embedding of the input prompt and $\epsilon\_{\boldsymbol{\psi}}$ is the output of our SpND model. In this way, the spherical net can extract panoramic features that guide the pre-trained diffusion model for panorama outpainting.

---

### Official Review · Reviewer_QS7j · 2025-03-17

**Overall Recommendation:** 3

**Summary:**

This work proposes to impose the sphere nature in the design of the diffusion model, such that the panoramic format is intrinsically ensured during the learning procedure, named the spherical-nested diffusion (SpND) model. In particular, the authors design to employ spherical noise in the diffusion process to address the structural prior, together with a newly proposed spherical deformable convolution (SDC) module to intrinsically learn the panoramic knowledge. Experimental results demonstrate the effectiveness of the proposed methods on various datasets.

**Claims And Evidence:**

See the weaknesses for more details.

**Essential References Not Discussed:**

See the weaknesses for more details.

**Experimental Designs Or Analyses:**

See the weaknesses for more details.

**Methods And Evaluation Criteria:**

See the weaknesses for more details.

**Other Comments Or Suggestions:**

The ablation study lacks the important qualitative evaluations.

**Other Strengths And Weaknesses:**

Strengths:
- The paper is generally well-structured, with illustrative diagrams effectively clarifying key concepts and model architectures. The motivation, design choices, and experiments are clearly presented, aiding reader comprehension.
- By directly incorporating spherical constraints into the diffusion process, the method improves quantitative metrics (FID) compared to existing state-of-the-art approaches.

Weaknesses:
- Limited Novelty of the SDC Layer Compared to Existing Panoramic Vision Approaches: The paper proposes a Spherical Deformable Convolution (SDC) layer, aiming to better capture the spherical nature of panoramic images. Yet, the technical design closely resembles prior methods proposed in existing panoramic vision literature. For example, "Eliminating the blind spot: Adapting 3d object detection and monocular depth estimation to 360 panoramic imagery", "Panoformer: panorama transformer for indoor 360° depth estimation", "Cylin-Painting: Seamless 360 panoramic image outpainting and beyond", "Panoramic panoptic segmentation: Insights into surrounding parsing for mobile agents via unsupervised contrastive learning", etc. Clarifying how the modifications substantially differ from the above approaches through more explicit analyses and discussion would significantly strengthen the novelty claim.
- Lack of Quantitative Evaluations on Non-i.i.d. Noise in ERP Format with Different Densities: The authors introduce the notion of non-i.i.d. spherical noise after ERP projection as a key component in their model. However, the paper does not quantitatively analyze how varying the sampling density impacts the performance or characteristics of the generated panoramic images. Although visual examples and qualitative insights are provided, rigorous numerical evaluations illustrating the effect of different densities and their impact on model performance (e.g., through ablation studies on various values) are missing. Such quantitative evaluations are crucial for understanding the practical robustness and sensitivity of the proposed method.
- Limited Capability Regarding Input Versatility (Multiple Images Support): The proposed SpND model primarily focuses on scenarios where the panoramic image outpainting input is a single masked ERP image. However, many realistic applications, particularly those involving panoramic image reconstruction in AR/VR or autonomous driving scenarios, require handling multiple overlapping or partial-view inputs simultaneously. The current formulation and experimentation do not clearly address or demonstrate whether or how the proposed approach could effectively scale to or support scenarios involving multiple input images or viewpoints. Such limitations constrain the practicality and applicability of the method to real-world panoramic image generation tasks.

**Questions For Authors:**

See above.

**Relation To Broader Scientific Literature:**

It is highly related to the panoramic image outpainting literature.

**Theoretical Claims:**

See the weaknesses for more details.

---

> ### Author Rebuttal · Authors · 2025-04-01
>
> Many thanks for the valuable comments!
> - **Tab.1 - 4 are in https://anonymous.4open.science/api/repo/tabl/file/T.pdf?v=5327f2ef**
> - **Fig.1 - 9 are in https://anonymous.4open.science/api/repo/figu/file/F.pdf?v=2d1be253**
>
> > ### **1 Weakness 1: Novelty of Our SDC Layer**
>
> To the best of our knowledge, the novelty of our SDC layer lies in the first successful attempt of proposing a new **basic convolution operation** to address the panoramic deformation, the key to achieve high-quality panoramic outpainting. Our spherical noise setting also contributes to our overall novelty in addressing this task. In contrast, existing methods, including those for other tasks such as panoramic image segmentation and depth estimation, typically address the panoramic deformation via soft loss regularization or feature re-sampling strategies. More specifically, Ref. [1] developed existing rectilinear architectures via pre-defined rectilinear-panoramic projection, to achieve 3D object detection and monocular depth estimation. Ref. [2] employed spherical resampling solely for the value feature within Transformer, to enhance panoramic image depth estimation. Ref. [3] added learnable positional encoding cues with existing features that are still processed by standard 2D convolution, for panoramic image outpainting. Ref. [4] regularized the network by spherical contrastive loss to effectively capture panoramic content for the segmentation of 360° images. However, Refs. [1, 2, 4] are incapable of generating panoramic content and thus unsuitable for the panoramic image outpainting task. More importantly, all the existing methods rely on either extra feature re-organization (e.g., Refs [2, 3]) or sphere-assisted losses (e.g., Refs. [1, 4])  to accommodate panoramic images. In contrast, our SDC layer essentially alters the basic convolution operations, which is fundamentally different from the existing strategies adopted for both outpainting and other panoramic-related tasks. Indeed, the fundamental improvement of our SDC layer on the convolution operation is much effective and straightforward to enforce geometry cues in addressing panoramic deformation, thus benefiting remarkable improved quality of outpainted images.
>
> Ref. [1]: Eliminating the blind spot: Adapting 3D object detection and monocular depth estimation to 360 panoramic imagery.
>
> Ref. [2]: Panoformer: Panorama transformer for indoor 360° depth estimation.
>
> Ref. [3]: Cylin-Painting: Seamless 360 panoramic image outpainting and beyond.
>
> Ref. [4]: Panoramic panoptic segmentation: Insights into surrounding parsing for mobile agents via unsupervised contrastive learning.
>
> > ### **2 Weakness 2: Quantitative Evaluations on Non-*i.i.d.* Noise in ERP Format**
>
> Yes. Our Non-*i.i.d.* noise setting caters for the panoramic format during the outpainting, and we further conducted quantitative evaluations as suggested by the reviewer. More specifically, we further ablated two settings by varying sample density $\zeta$, i.e., via $\zeta=15$ and $\zeta=45$, and report the results in Tab. 1 in the link. As shown in the table, our SpND method achieves optimal performance at $\zeta = 45$ and $\zeta = 30$, consistent with the analysis in Supplementary-A, where the ERP noise closely aligns with  the ground-truth features/images. Compared to $\zeta = 45$, $\zeta = 30$ offers the highest efficiency. Choosing $\zeta = 15$ slightly degrades outpainting performance but still outperforms existing baselines. These results confirm the rationale and robustness of our proposed non-*i.i.d.* ERP noise, derived from *i.i.d.* spherical noise.
>
> > ### **3 Weakness 3: Limited Capability Regarding Input Versatility**
>
> We agree with the reviewer that outpainting from multiple input views can further improve the practicality and applicability of our method. Indeed, our model can readily accommodate multiple partial-view and overlapping inputs by adjusting the training masks accordingly. More specifically,
>
> - For the multi-input scenario, we newly evaluated our SpND method based on two viewpoints centered at $(\theta_1=-90^\circ, \phi_1=-15^\circ)$ and $(\theta_2=90^\circ, \phi_2=15^\circ)$ within the equirectangular projection (ERP) format, thus denoted as  **Dual**.
> - For the overlapping scenario, we newly evaluated our SpND method based on two viewpoints centered at $(\theta_1=20^\circ, \phi_1=0^\circ)$ and $(\theta_2=90^\circ, \phi_2=15^\circ)$ with overlapped regions, also using the ERP format. We denote this as **Overlapping**.
>
> We report the results in Tab. 2, together with the subjective results in Figs. 1-2 in the link. From this table and figure, we can conclude that our method can generalize to multiple overlapping and partial-view inputs. Note that due to limited rebuttal period, the model was trained by adequate convergence, and performance improvements can be expected with further training. Even though, our SpND method still achieves superior performances for the panoramic image outpainting task.

---

> > ### Comment · Reviewer_QS7j · 2025-04-05
> >
> > Thanks for your detailed response, which addressed most of my concerns well. Thus, I would like to increase my rating. However, the revision parts, especially the clarifications of the novelty compared with previous panoramic works (regarding the weakness 1), are expected to be presented in the paper. I believe such a clarification is crucial to highlight the contribution of this work.

---

> > > ### Author Response · Authors · 2025-04-05
> > >
> > > Thank you for the positive opinions and valuable suggestions. It is really great that you would like to increase your rating. Yes. We totally agree with that clarifying our difference against existing panoramic methods is important to highlight our novelty and contributions, especially regarding our SDC layer. During the rebuttal period, we are not allowed to submitted a revised version. In our final revised paper, we shall comprehensively discuss the suggested 4 references in detail, by including our response to Weakness 1 with further improved clarifications, as well as trying to find possible other relevant references.

---

### Decision · Program_Chairs · 2025-05-01

**Decision:**

Accept (poster)

**Comment:**

This paper presents a well-motivated and technically sound contribution to panoramic image generation that gradually becomes a popular research topic. Specifically, the proposed spherical noise and deformable convolution can benefit the panoramic images' ERP distortions. Through rebuttal efforts, all the reviewers positively commented on the submission and suggested that the authors clarify and emphasize distinctions from prior works in the final revision.

After reading the paper and rebuttal, the AC votes for weak acceptance of its novel architecture design, extensive experimental results, and impact on the specific area.

### Method

By introducing spherical noise and a spherical deformable convolution (SDC) layer, the model intrinsically captures the spherical geometry of panoramic images. Reviewers appreciated the sound motivation and architectural clarity.

### Experiment

SpND outperforms state-of-the-art baselines such as PanoDiffusion and AOG-Net, demonstrating significant improvements in FID and FAED across multiple datasets. Both qualitative and quantitative results are convincing.

###  Rebuttal Period

Authors addressed all reviewer concerns with additional experiments, ablation studies, visualizations, and theoretical clarifications, leading to improved reviewer ratings.